# CENTROID-BASED LEARNING FOR MALWARE DETECTION AND NOVEL FAMILY IDENTIFICATION

## ABSTRACT

Detecting out-of-distribution (OOD) data categories while preserving the accuracy of existing classifications is a pressing challenge in many domains. Conventional methods often falter when tasked with generating or identifying new data classes, especially when dealing with graphical data and the problem of graph isomorphism. In this paper, we present a novel approach, the Graph Centroid Model (GCM), which combines Control Flow Graphs (CFGs) with a Graph Neural Network (GNN) to address this challenge effectively. The GCM assigns embeddings produced by a GNN to partitions that support the classification of both known and new classes, even those absent during training.

Our approach quantifies the differences between samples in the embedding space, enabling the identification of multiple distinct representations of familiar classes during training while providing a straightforward mechanism for detecting new classes during testing. This not only improves classification accuracy but also offers intuitive visualizations that provide valuable insights. When applied to a benchmark malware dataset (BODMAS), our method reveals structural commonalities among samples from different malware families while effectively discerning new, previously unseen classes based on their distance from learned representatives in the embedding space.

## 1 INTRODUCTION

Traditional techniques for detecting new malware families, especially emerging variants, have long relied on labor-intensive feature engineering and domain-specific knowledge (Gasmi et al., 2019). While Control Flow Graphs (CFGs) have demonstrated potential in malware classification (Gao et al., 2022a), their full capabilities are yet to be fully explored, especially regarding the detection of out-of-distribution (OOD) new-family threats and the ever-changing landscape of evolving malware.

Our work addresses these needs by introducing a novoel *Graph Centroid Model* (GCM), which we use to learn a representation over embeddings produced by a Graph Neural Network (GNN) that can be used both to effectively classify known families of malware, and to identify new families as they emerge at test time. In particular, the representation learned by the centroid model locates class representatives precisely in the embedding space, providing a straightforward way to evaluate new samples in relation to these representatives for the purpose of classification. Additionally, the distance between new data points and existing class representatives provides a basis for determining whether a data point belongs to a known class or whether it represents a class that was not known during training.

We present a comprehensive evaluation of our method on the BODMAS dataset of x86 malware executables (Yang et al., 2021). The results of this study demonstrate that our approach brings key advantages.

- *Effective in Supervised Settings*: Our approach excels in supervised learning scenarios, classifying malware samples from known families with precision and accuracy. Leveraging embeddings derived from malware CFGs, our model uncovers semantically rich features from the data, obviating the need for manual feature engineering. These features empower accurate classification of known categories, consistently outperforming traditional techniques that rely on hand-crafted features.

- *Effective in Novel Class Settings*: Crucially, our method extends its effectiveness to novel class samples, showcasing its ability to classify new malware families as novel categories. This unsupervised capability rests on the same fundamental insight: the symbiotic relationship between CFGs and GNNs enables our model to unveil intricate patterns in data, even when labeled categories are absent. It excels in identifying new, potentially emerging, or hitherto undiscovered trends and patterns in malware, making it an *invaluable tool for zero-day malware detection*.

In addition to our method, we contribute a novel dataset containing control flow graphs derived from the BODMAS dataset (Yang et al., 2021). This openly accessible resource enables researchers to delve into temporal analyses of PE malware from 2021, further enriching the cybersecurity community's arsenal.

Section 2 provides background on the problem, Section 3 elaborates on our methodology and dataset, Section 4 presents the experimental results, and Section 5 concludes this study. Through this research, we aspire to offer a robust solution for malware detection and classification, one that deftly tackles the ever-expanding challenge posed by new classes of malware in today's digital landscape.

## 2 RELATED WORK

There has been significant research on using machine learning techniques for malware detection. One approach is to use support vector machines (SVM) to identify behavioral changes within a malware family (Wadkar et al., 2020). Another approach is to train models on a dataset of benign PE files and use the entire raw binary as input to detect malicious samples (Raff et al., 2017). Visual similarity among malware instances can also be used for classification using standard image features (Nataraj et al., 2011). A combination of convolutions followed by long short-term memory (LSTM) is used to process the sequence of application programming interface (API) calls generated by malware files (Kolosnjaji et al., 2016).

Deep learning techniques, such as convolutional neural networks (CNNs), have also been used for malware classification by converting malware samples into grayscale images (Hsiao et al., 2019). However, these models are not robust to adversarial perturbations, as demonstrated by the success of adversarial examples in evading machine learning models (Spencer et al., 2022). This work spurred an entire field of adversarial robustness for malware detection using the raw bytes, including much work that uses CNNs on the bytes. Papers like Suciu et al. (2019) demonstrate architectural weaknesses of using convolutions on the bytes.

New malware family discovery belongs to the general setup of detection and classifying Out-Of-Distribution (OOD) samples in the machine learning community. Malware detection, unlike standard object detection or classification tasks, can significantly benefit from identifying unseen classes that are out of training distribution. They, especially modern variants, often exhibits complex and polymorphic behaviors, designed precisely to evade traditional signature-based detection methods. These behaviors manifest as intricate patterns within the control flow graph, where certain sequences of instructions or code structures are indicative of malicious intent.

### 2.1 BACKGROUND ON OOD DETECTION

Detecting and classifying Out-of-Distribution (OOD) samples is crucial, especially in applications like anomaly detection, fraud detection, and image recognition. Various techniques have been proposed to address this challenge, leveraging different approaches and methodologies.

Probabilistic Modeling: Probabilistic modeling methods aim to estimate the uncertainty associated with predictions. Bayesian Neural Networks (BNNs) (Silvestro & Andermann, 2020) and Monte Carlo Dropout (Miok et al., 2019) are popular techniques for estimating uncertainty in neural networks. By modeling the predictive uncertainty, these methods can detect OOD samples by flagging instances with high uncertainty.

Feature Space Distributions: Techniques based on feature space distributions assess the distribution of feature representations of data. Mahalanobis Distance (Denouden et al., 2018) (Xu et al., 2020) measures the Mahalanobis distance of a sample's feature vector from the distribution of in-distribution samples. If the distance exceeds a threshold, the sample is considered OOD. Similarly,

methods using Gaussian Mixture Models (GMM) (Terejanu et al., 2008) estimate the feature space distribution and classify samples based on likelihood.

Deep Generative Models: Deep Generative Models, such as Variational Autoencoders (VAEs) (Doersch, 2016) and Generative Adversarial Networks (GANs) (Goodfellow et al., 2020), can be used for OOD detection. These models learn to generate data that resembles the training data distribution. OOD samples may have a higher reconstruction error or be less likely to be generated by the model, making them distinguishable.

Ensemble Methods: Ensemble methods combine multiple models to improve OOD detection. Techniques like ODIN (Liang et al., 2017) enhance the confidence score of the model by applying a temperature scaling and adversarial perturbations. By combining multiple models and their predictions, ensemble methods often achieve better OOD detection performance. (Choi et al., 2018)

Open Set Recognition: Open Set Recognition methods explicitly address the problem of OOD detection. One-Class SVM (Li et al., 2003) and Isolation Forests (Liu et al., 2008) are classical techniques for open set recognition. These methods model the in-distribution samples and classify samples as in-distribution or OOD based on their proximity to the model.

## 2.2 GRAPHICAL DATA FOR OOD MALWARE DETECTION

Control flow graphs can reveal potential vulnerabilities or hidden backdoors within the code. This information is essential for assessing the malware's potential impact and behavior (Wu et al., 2021) (Jha et al., 2013). CFGs empower analysts to dissect malware, enhance threat detection, and improve software quality through their visual representations and insights into complex code structures. These techniques significantly contribute to the analysis, mitigation, and overall enhancement of software security. (Gao et al., 2022b) Prior work has used GCNs on Function Call Graphs to provide robust malware classification (Kargarnovin et al., 2021) (Yan et al., 2019).

Previous research has proposed several approaches for dealing with clustering uncertainty in the context of malware analysis. For example, some methods use ensemble clustering to combine multiple clusterings and increase robustness to noise. Other methods use hierarchical clustering to build a tree-like structure of clusters, which can capture the hierarchical relationships between different families. (Kinable & Kostakis, 2010) (Bayer et al.)

Despite these advances, discovering new malware families remains a challenging problem. In this paper, we propose a new approach that leverages the uncertainty of partitioning to identify previously unknown families. A centroid model is a type of unsupervised learning model that learns a set of centroids or prototypes for a given dataset. Given a new input, the model assigns it to the centroid that is closest in terms of distance. Centroid models have been used in many domains, including text classification, image classification, and anomaly detection.

## 3 METHODS

### 3.1 DATASET

We use a subset of the BODMAS: An Open Dataset for Learning based Temporal Analysis of PE Malware for our experimentation (Yang et al., 2021).

To generate CFGs for the malware samples in the BODMAS dataset, we employed the Binary Analysis Platform (BAP) (Brumley et al., 2011). BAP loads the raw byte files, parses the binary instructions, and constructs an intermediate language (IL) of the binary program. For each instruction, BAP constructs a node within the CFG. These nodes collectively form the CFG, where edges between nodes represent the flow of control between instructions or basic blocks. In essence, the CFG provides a visual representation of the malware's execution flow, enabling us to gain insights into its behavior and characteristics. The Linear Sweep Disassembly Algorithm is a fundamental technique used by BAP for converting raw binary code into a structured CFG. The algorithm relies on the following key principles:

The algorithm scans through the binary code sequentially, identifying instructions. An instruction is denoted as 'I' and typically consists of an opcode and operands. This can be expressed mathemati-

cally as:

$$I_i = (\text{opcode}_i, \text{operands}_i)$$

BAP then identifies control flow instructions that alter the program's execution path, such as branches, jumps, and calls. These instructions are critical for CFG construction and are represented as 'CFI'.

$$CFI_i = \begin{cases} 1 & \text{if } I_i \text{ is a control flow instruction} \\ 0 & \text{otherwise} \end{cases}$$

Basic blocks are sequences of instructions with a single entry point and a single exit point. They are defined as:

$$BB_i = (I_{\text{entry}_i}, I_{\text{exit}_i})$$

The CFG is built by linking basic blocks through control flow instructions. Each basic block is represented as a node in the graph, and control flow instructions create edges between these nodes.

This algorithm ensures that the binary code is systematically disassembled into basic blocks and that control flow between these blocks is accurately captured in the CFG.

## 3.2 CENTROIDS

Centroid Nets offer a fundamental shift in the way we approach classification tasks. They provide a versatile tool for various machine learning tasks by combining a neural network $f : \mathbb{R}^d \to \mathbb{R}^m$ with a set of centroids $\{c_i, y_i\}_{i=1}^n$. These centroids $c_i \in \mathbb{R}^m$ represent concrete locations in an embedding space, and $y_i \in [C]$ assigns labels to these centroids. The Centroid Net makes predictions for input $x$ by determining the closest centroid using the Euclidean distance:

$$F(x) = y_{i^*}, \quad i^* = \arg \min_j ||f(x) - c_j||. \tag{1}$$

This approach generalizes traditional classification methods, offering several distinct advantages.

Firstly, Centroid Nets eliminate the need for manually crafting representative samples for each class. In traditional classification, it is often necessary to handpick a few samples to represent a class, a process that can be subjective and prone to bias. With Centroid Nets, the network itself learns where these representatives should be located in the embedding space based on the data it's exposed to. This not only simplifies the workflow but also provides more data-driven and principled class representatives.

Secondly, in traditional methods, inputs are classified based on log-likelihood scores or probabilities, which don't inherently capture the structure of the data. Centroid Nets, on the other hand, provide concrete locations for each class's representative. This knowledge is invaluable. It enables us to evaluate new samples not just in terms of their class label but also in relation to these representatives. This is important for two main reasons:

**New Family Detection**: OOD detection is a ubiquitous challenge in various domains, including malware classification. With Centroid Nets, this task becomes more intuitive. By comparing the distances between new data points and existing class centroids, we can make informed decisions about whether a data point belongs to a known class or represents a new and potentially unknown category. This capability adds a natural mechanism for general OOD detection.

**Interpretability**: The Centroid Net's approach provides interpretable results. When making a classification decision, we not only know the predicted label but also the closest centroid. This proximity-based explanation enhances our understanding of why a particular classification decision was made. It allows us to explain decisions by finding nearby representatives, providing a more intuitive grasp of model behavior.

A Centroid Net is defined as a neural network $f : \mathbb{R}^d \to \mathbb{R}^m$ along with a set of centroids $\{c_i, y_i\}_{i=1}^n$. Here, $c_i \in \mathbb{R}^m$ represents centroid coordinates, and $y_i \in [C]$ represents labels. The Centroid Net predicts the label for input $x$ based on the closest centroid using the Euclidean distance:

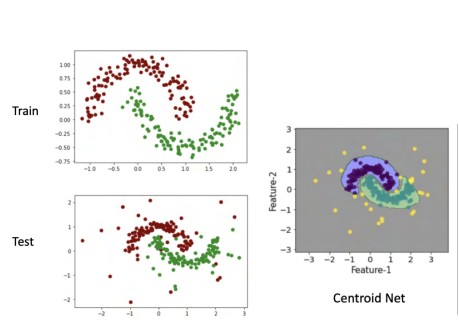

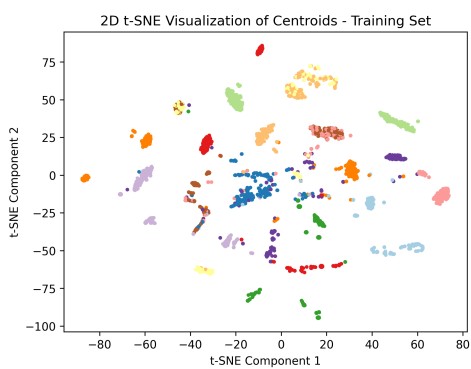

(a) Decision Boundaries of a Centroid Net (with Rejection)

(b) t-SNE Projection of Centroid Distances (Training Set)

Figure 1: Visualization of Centroid Net Abilities

$$F(x) = y_{i^*}, \quad i^* = \arg\min_j \|f(x) - c_j\|. \tag{2}$$

The prediction $F(x)$ is determined by finding the centroid that minimizes the Euclidean distance between $f(x)$ and each centroid $c_j$:

$$F(x) = y_{i^*}, \quad i^* = \arg\min_j \|f(x) - c_j\|. \tag{3}$$

In practice, this assigns inputs to centroids in feature space, facilitating effective classification.

Centroid Nets offer instance-based explanations. The nearest centroid represents the prototype input concerning the input of interest. We compute the prototype $p_i$ of a centroid $c_i$ through gradient descent, providing a measure of similarity between the input and the prototype:

$$p_i = \arg\min_p \|f(p) - c_i\|. \tag{4}$$

The difference between the input and its prototype can be viewed as *similarity*: $(x, p) = p - x$.

Figure 1a provides plots of decision boundaries of a Centroid Net (with rejection) trained on TwoMoons. Yellow points are labeled as $\perp$ by the network, i.e. rejection from prediction. The decision boundary is closed for Centroid Models, whereas linear models would have an open decision boundary. Figure 1b illustrates the ability of our method to separate malware families. Each color represents a different malware family, and suggests inter-class structural similarities that are represented in embedding space.

## 3.3 Centroid Nets within Neural Networks

Centroid Nets represent a fundamental innovation in neural network architecture, providing a versatile tool for various machine learning tasks. These networks are defined as $f : \mathbb{R}^d \to \mathbb{R}^m$ along with a set of centroids $\{c_i, y_i\}_{i=1}^n$, where $c_i \in \mathbb{R}^m$ represents centroid coordinates, and $y_i \in [C]$ represents labels. The Centroid Net makes predictions for input $x$ by determining the closest centroid using the Euclidean distance:

$$F(x) = y_{i^*}, \quad i^* = \arg\min_j \|f(x) - c_j\|. \tag{5}$$

The significance of the Centroid Net lies in its ability to create concrete locations in an embedding space, essentially providing reference points for each class. This is in contrast to conventional

methods where you need to manually create representatives for each class. The Centroid Net offers a more principled and data-driven approach.

This characteristic transforms the task of classification. With Centroid Nets, we not only perform classification but also gain a unique perspective on our data. We precisely know where class representatives are located in the embedding space. This knowledge enables you to evaluate new samples in relation to these representatives, a process that naturally lends itself to out-of-distribution (OOD) detection.

### 3.4 Comprehensive Model Structure

Our classification and discovery model comprise a Graph Convolutional Network (GCN) integrated with Centroid Models. This architecture, augmented with batch normalization and global pooling, is designed for the efficient detection of known malware families and the exploration of potential new families.

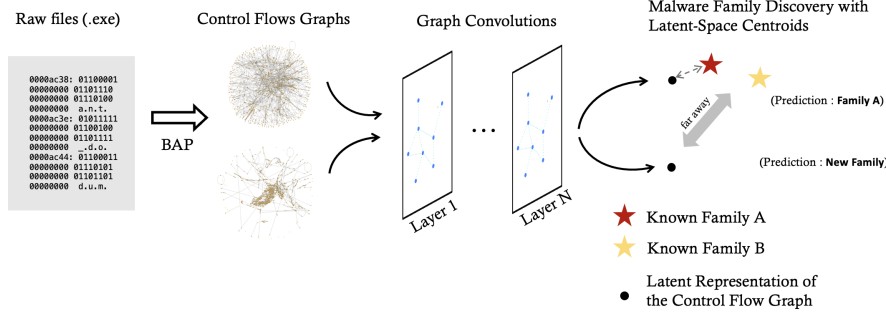

Figure 2: Method: Executable files are transformed into Control Flow Graphs (CFGs), enabling the application of a Graph Convolutional Network (GCN) with a centroid layer for new malware class discovery.

Our approach to classification and discovery hinges on specific rules based on the epsilon threshold:

- Single Output Label: If the margin between distances from the input to class centroids is greater than $\epsilon$, a single output label $y_i$ is assigned.
- Abstention: When the input is far from all class centroids, leading to an unlikely match with any trained classes, an abstention $\perp$ is triggered.

In practice, $\epsilon$ is chosen based on the tradeoffs between precision and recall. Smaller values of epsilon prioritize precision by accepting only data points very close to centroids, potentially at the cost of recall. Conversely, larger values of epsilon prioritize recall by accepting a broader range of data points, potentially at the cost of precision. The method by which centroids are selected is described in more detail in A.

### 4 Results

In the context of malware detection, we work with multiple sets of x86 executable files, each associated with a distinct family of malware. Let $F$ be the set of all families, and for each family $f \in F$, there exists a corresponding set $P_f$ of programs. Additionally, we associate each family $f$ with a set of signatures $M_{s_f}$. A malware detector, denoted as $D$, operates on the union of all program sets $\bigcup_{f \in F} P_f$ and the union of all signature sets $\bigcup_{f \in F} M_{s_f}$.

To assess the performance of a malware detector, we distinguish between programs that belong to different families. For this purpose, we assume that the intersection between any two program sets $P_f$ and $P_{f'}$ for $f \neq f'$ is empty, and the same holds for the intersections between any two signature sets $M_{s_f}$ and $M_{s_{f'}}$.

## 4.1 TRAINING PROCESS

We train the GCN Centroid model and benchmarks (GCN and GraphSage) with 1000 epochs. Figure 3) shows the training loss and accuracy. GCNCentroid converges fast within 200 epochs and the training loss stays at 0.36, which prevents the model from overfitting and reaches higher training accuracy than GCN and GraphSage.

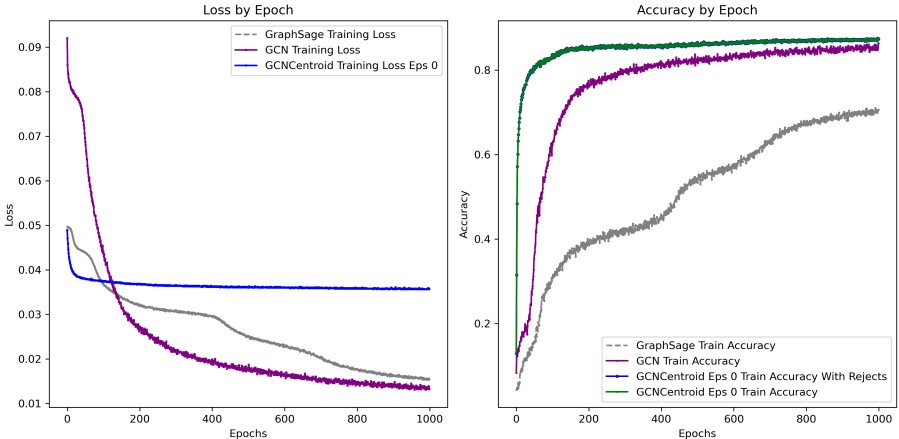

Figure 3: Train Loss and Accuracy over 1000 epochs versus benchmarks

We then train our GCN Centroid model over various values of epsilon and record training loss and accuracy (Figure 4). As shown below, epsilon of 0 has the highest accuracy for training with rejection, however there is a trade-off with new family detection.

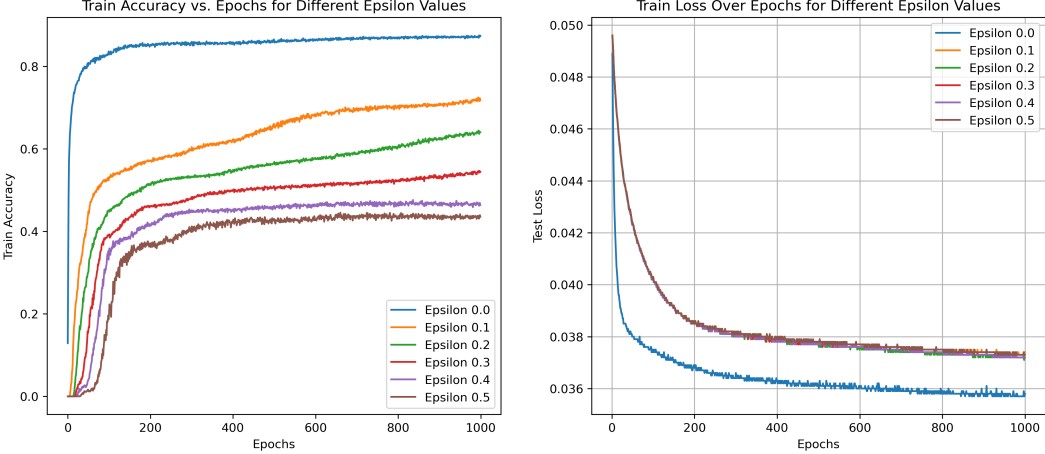

Figure 4: Train accuracy and loss over 1000 epochs under different values of epsilon

## 4.2 TRADEOFFS OF NEW FAMILY DETECTION

By comparing the test performance of detecting existing classes and new families, as shown in Table 1, when evaluated on the training set, all models exhibit good precision and recall, with the GCNCentroid at any $\epsilon$ value with the highest precision compared to GCN and GraphSage.

The introduction of the rejection mechanism significantly affects model performance. GCN and GraphSage cannot handle rejections and are thus omitted from this evaluation. Among the models that support rejection, GCNCentroid ($\epsilon = 0.1$) stands out with a precision of 81.65% and recall of

| | Train | | Test No Reject | | Test With Reject | | New Family |
|---|---|---|---|---|---|---|---|
| | Precision | Recall | Precision | Recall | Precision | Recall | Accuracy |
| GCN | 0.8344 | 0.8349 | 0.9083 | 0.8948 | - | - | 0.0583 |
| GraphSage | 0.8014 | 0.8023 | 0.8080 | 0.8115 | - | - | 0.0250 |
| GCNCentroid ($\epsilon = 0$) | 0.8929 | 0.8729 | 0.8572 | 0.8448 | 0.8572 | 0.8448 | 0 |
| GCNCentroid ($\epsilon = 0.1$) | 0.8928 | 0.8745 | 0.8711 | 0.8906 | 0.8165 | 0.6844 | 0.7500 |
| GCNCentroid ($\epsilon = 0.2$) | 0.8839 | 0.8711 | 0.8842 | 0.8771 | 0.7895 | 0.6344 | 0.9417 |
| GCNCentroid ($\epsilon = 0.3$) | 0.8975 | 0.8779 | 0.8441 | 0.8479 | 0.4167 | 0.3563 | 0.9917 |
| GCNCentroid ($\epsilon = 0.4$) | 0.8757 | 0.8661 | 0.8626 | 0.8688 | 0.2500 | 0.1802 | 1.0000 |
| GCNCentroid ($\epsilon = 0.5$) | 0.8938 | 0.8779 | 0.8755 | 0.8875 | 0.1667 | 0.1667 | 1.0000 |

Table 1: Comparison with Baselines and Different Values of $\epsilon$. GCN and GraphSage are unable to detect new families. Difference epsilon provides tradeoffs in detecting existing and new families.

68.44%. This balance allows it to reject uncertain samples while maintaining a reasonable level of correct classifications.

Different scenarios demand different trade-offs between precision and recall. GCNCentroid ($\epsilon = 0.3$) achieves a balanced performance on the test set without rejection, making it suitable for scenarios where both false positives and false negatives must be minimized. Handling Rejection: Models like GCNCentroid ($\epsilon = 0.1$) excel in situations where a rejection mechanism is crucial, allowing the model to decline uncertain samples effectively.

New Family Detection: GCNCentroid ($\epsilon = 0$) and GCNCentroid ($\epsilon = 0.1$) exhibit high accuracy in detecting new malware families, a critical capability in the evolving landscape of cybersecurity.

## 4.3 CENTROIDS AID ABILITY TO VISUALIZE

The t-SNE plots (Figures 5) reveal a stark separation of centroids into well-defined partitions. Each partition represents a distinct grouping of centroids based on their high-dimensional feature representations. Notably, these partitions exhibit minimal overlap, signifying the effectiveness of the t-SNE technique in capturing subtle relationships within the data.

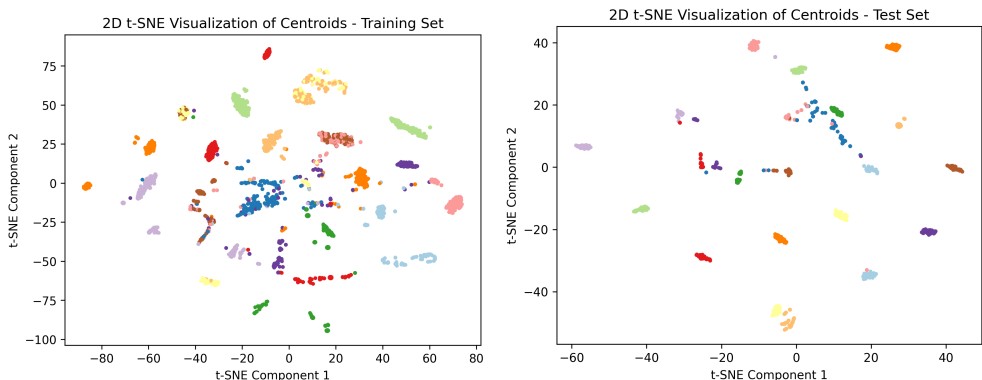

Figure 5: t-Distributed Stochastic Neighbor Embedding (t-SNE) Projection of Centroid Distances of Training Set and Test Set.

The distinctiveness of these partitions suggests that the centroids share common structural characteristics that enable their segregation into homogeneous groups. Such a clear demarcation is valuable for understanding the underlying diversity and organization of the centroids, facilitating their categorization into meaningful families or classes.

Notably, some families are mapped directly on top of other families. This suggests that malware family labels created by domain experts may have structural similarities indicating that one family directly evolved from an earlier family.

Furthermore, the t-SNE plot provides an insightful visualization of centroid relationships that may not be readily apparent in higher-dimensional spaces. This visualization aids in both qualitative analysis and the potential development of novel methods for centroid classification and discovery. Figure 6 visualizes the embedding of the three new malware families. Interestingly, the three new families are plotted close together in the embedding space.

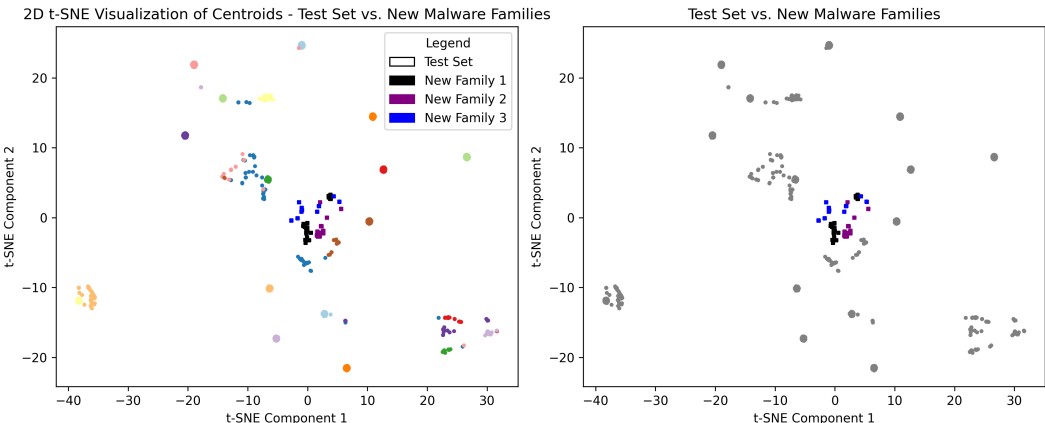

Figure 6: t-Distributed Stochastic Neighbor Embedding (t-SNE) Projection of Centroid Distances of Test Set versus New Malware Families.

## 5 DISCUSSION AND CONCLUSION

In this work, we have introduced an approach to the challenges of out-of-distribution (OOD) detection and malware classification. We use a Centroid Net architecture, a dynamic framework that classifies both new and known threats.

**Adaptability in the Face of Change**: The Centroid Net is not a static entity but a dynamic, malleable component that integrates within neural network architectures. Through optimization and training with standard classification losses, the Centroid Net evolves alongside the neural net in which it is embedded, continually refining its centroids and feature representations. It therefore provides a suitable tool for changing distributions of data, because of its ability to distinguish between new samples that are near class representatives and those that are not.

**Concrete Locations in the Embedding Space**: Unlike traditional classifiers that often require manual intervention to define class representatives, the Centroid Net creates clear, data-driven locations in an embedding space. This equips us with a precise understanding of where class representatives reside in the vast feature space, allowing for interpretability, user analysis, and visualization.

**A Natural Path to OOD Detection**: The Centroid Net is inherently capable of facilitating OOD detection. By computing the distances between new data points and existing class centroids, we can discern with confidence whether a data point belongs to a known family or represents an entirely new and unfamiliar category. We showed that we were able to do this with very high precision and recall, compared to classic graph network techniques.

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

# A  APPENDIX

## A.1  ALGORITHM OF CENTROID PARTITIONING CALCULATION

---

**Algorithm 1** Centroid Partitioning for Malware Family Discovery

---

0: **procedure** CENTROIDPARTITIONING(samples, threshold)
0:   $centroids \leftarrow$ RANDOMSAMPLE($samples, k = 1$) {Choose a random sample as the first centroid}
0:   **for** $sample$ in $samples$ **do**
0:     $distances \leftarrow$ COMPUTEDISTANCES($sample, centroids$) {Compute distances to all centroids}
0:     $min\_distance \leftarrow$ MIN($distances$) {Find minimum distance}
0:     **if** $min\_distance > threshold$ **then**
0:       $centroids \leftarrow$ APPEND($sample$) {Add sample as a new centroid}
0:     **end if**
0:   **end for**
0:   **return** $centroids$
0: **end procedure**

0: **procedure** COMPUTEDISTANCES(sample, centroids)
0:   $distances \leftarrow []$
0:   **for** $centroid$ in $centroids$ **do**
0:     $distance \leftarrow$ DISTANCE($sample, centroid$) {Compute distance between sample and centroid}
0:     $distances \leftarrow$ APPEND($distance$) {Add distance to list of distances}
0:   **end for**
0:   **return** $distances$
0: **end procedure**=0

---

The CentroidPartitioning procedure takes in a set of samples and a threshold distance value. It starts by randomly selecting one sample from the set as the first centroid. It then iterates over each sample and computes the distance to all existing centroids using the ComputeDistances procedure. If the minimum distance to any centroid is greater than the threshold, the sample is added as a new centroid.

The ComputeDistances procedure takes in a sample and a set of centroids. It iterates over each centroid and computes the distance between the sample and the centroid using a distance metric such as Euclidean distance or Manhattan distance. It returns a list of distances between the sample and each centroid.

## A.2  FUTURE WORK

In the context of malware detection and classification, the persistent challenge of graph isomorphism has been widely acknowledged. Isomorphic graphs, possessing identical structures while representing different malware samples, present a formidable obstacle to traditional analysis methods. In our future work we will explore strategies to address graph isomorphisms. To enhance the robustness of our method, we intend to explore ensemble techniques that amalgamate predictions from multiple models. In particular, we aim to leverage community detection algorithms to pinpoint 'suspicious' subgraphs within larger graphs, potentially revealing the malware components within each program. We propose evaluating the effectiveness of our approach in handling graph isomorphism using established metrics such as the Adjusted Rand Index (ARI) and Normalized Mutual Information (NMI). These metrics may provide empirical evidence of our method's ability to robustly partition isomorphic substructures within malware families.

Control flow graphs (CFGs) are generated using two primary analysis techniques: static analysis and dynamic analysis. Static analysis involves scrutinizing the malware's source code or binary without its execution, offering insights at the code level. In contrast, dynamic analysis entails executing

the malware in a controlled environment to observe its runtime behavior, which is crucial for understanding its actions. Other future work will focus on optimizing the synergy between static and dynamic analysis. Combining insights from both approaches will provide a more comprehensive understanding of malware functionality. We aim to refine this integration to capture a wider range of malware behaviors, enhancing our ability to classify and detect previously unseen threats.

