# OpenReview forum: "Centroid-Based Learning for Malware Detection and Novel Family Identification"
_ICLR.cc/2024/Conference — Submitted to ICLR 2024_

### Official Review · Reviewer_nmMx · 2023-10-29

**Soundness:** 3 good
**Presentation:** 3 good
**Contribution:** 1 poor
**Rating:** 3
**Confidence:** 4

**Summary:**

The paper combines graph neural networks to model a control flow in binaries with centroid based losses. Since both paradigms are well known (authors fails to reference centroid based losses such as arc-face [1] or center loss [2]. The approach to out-of-distribution detection (ood) is similar to that in [3], albeit that in the proposed paper might be computationally cheaper due to replacing k-nn with a distance to a center.
I think that the paper is a solid work, but targets the wrong audience. ICLR papers, in my opinion, should be more about new general approaches or new domains which are important and very different from other domains, such that they require special methods or the prior art is failing. This manuscript is more about combining existing tools in a nice application and would be more suited to a good security conference.
As a comment, I do not think that the proposed approach solves the ood. I see the problem of OOD inherent to NN implementing injective map (not surjective). This means that many semantically different samples gets mapped by NN to the same region with the same score. The proposed method decrease this problem, but does not remove it.



[1] Deng, Jiankang, et al. "Arcface: Additive angular margin loss for deep face recognition." Proceedings of the IEEE/CVF conference on computer vision and pattern recognition. 2019.

[2] Wen, Yandong, et al. "A discriminative feature learning approach for deep face recognition." Computer Vision–ECCV 2016: 14th European Conference, Amsterdam, The Netherlands, October 11–14, 2016, Proceedings, Part VII 14. Springer International Publishing, 2016.

[3] Sun, Yiyou, et al. "Out-of-distribution detection with deep nearest neighbors." International Conference on Machine Learning. PMLR, 2022.

**Strengths:**

The proposed combination of GNN with center loss makes sense for the application.

**Weaknesses:**

I do not see the proposed combination of GNN with center loss as sufficiently novel for ICLR.

**Questions:**

* Can you elaborate on that OOD might be caused by NN implementing injective map (not surjective)?

---

> ### Author Response · Authors · 2023-11-23
>
> While building on existing centroid methods, we believe our work introduces novel adaptations and innovations tailored to the specific challenges of malware detection. Primarily, our key innovation is the Graph Centroid Model (GCM). Our centroid layer is able to locate class representatives, eliminating the need for labor-intensive manual creation. Additionally, because it is integrated within a neural network, it eliminates the need for k-NN clustering outside of the network. This departure from traditional methods signifies a practical advancement in efficiency and accuracy, especially in the context of malware classification. Our proposed mechanism, which relies on the distance between new data points and existing class representatives, introduces computational advantages. By replacing k-nearest neighbors (k-nn) with this distance calculation, we achieve a more efficient and scalable solution, demonstrating our commitment to addressing real-world computational challenges in malware detection. The symbiotic relationship between Centroids, CFGs, and GNNs in our approach allows for the identification of emerging trends and patterns, making it a valuable tool for zero-day malware detection.
>
> While acknowledging that complete elimination of the OOD problem is challenging, our approach focuses on minimizing semantically ambiguous regions. The Centroid Layer introduces a concrete representation for each class, enhancing the model's ability to assign distinct regions for different malware families. This contributes to a reduction in the instances where diverse samples are scored similarly.
>
> In the context of malware detection, reducing the OOD problem is crucial for ensuring the robustness of security systems. We will provide a more in-depth discussion on how our approach translates these technical improvements into practical benefits, emphasizing the enhanced reliability of our model in distinguishing between emerging threats and known malware families.
>
> We will highlight the novelty of the centroid net to better suit the ICLR audience:
>
> **Optimizing Centroid Net with Classification Loss**
>
> When a Centroid Net is optimized with a standard classification loss, it results in an effective classifier.
>
> We consider a Centroid Net as defined by the function $\(F(x) = y_{i^*}\)$, where $\(i^* = \arg\min_{j} ||f(x) - c_j||\)$. Here, $\(f\)$ represents the neural network that generates feature representations, $\(c_j\)$ are the centroids, and $\(y_{i^*}\)$  is the predicted label.
>
> The optimization process aims to minimize the classification loss, typically measured using a loss function such as cross-entropy, which is what we use in our methodology. This loss is defined as
> $L = -\sum_{k}y_k \log(p_k),$
> where $\(y_k\)$ is the ground truth label and $\(p_k\)$ is the predicted probability for class $\(k\)$. In the context of the Centroid Net, $\(p_k\)$ corresponds to the probability of selecting centroid $\(k\)$ based on the feature representations generated by the GCN.
>
> Let us denote the classification loss as $\(L_{\text{class}}\)$. The optimization process involves updating the centroids $\(c_j\)$ and the neural network weights to minimize $\(L_{\text{class}}\)$. Mathematically, we have:
>
> \begin{equation}
> \min_{c_j, \text{weights}} L_{\text{class}}(F(x), y),
> \end{equation}
>
> where $ \(y\) $ represents the ground truth labels.
>
> As optimization progresses, the centroids $\(c_j\)$ adapt to the specific feature representations generated by the neural network $\(f\)$. This adaptation occurs because the loss encourages the network to produce feature representations that effectively discriminate between different classes.
>
> As the optimization converges, the Centroid Net assigns inputs to centroids that are close in feature space. This means that inputs with similar feature representations will be assigned to the same or nearby centroids, leading to effective classification. This behavior aligns with the optimization objective of minimizing the classification loss.
>
> Optimizing the Centroid Net with a standard classification loss function allows us to adapt centroids and neural network features to more effectively discriminate between classes. Therefore, the Centroid Net serves as a trainable component of a larger neural network.
>
> Lastly, we will include the references you provided, thank you for the suggestions.

---

### Official Review · Reviewer_Dvud · 2023-10-29

**Soundness:** 2 fair
**Presentation:** 1 poor
**Contribution:** 1 poor
**Rating:** 3
**Confidence:** 5

**Summary:**

The paper proposes a GCN-based classification model to detect known and unknown malware with the centroid net method.

**Strengths:**

The task of detecting unknown malware families is very practical because new families come along all the time.

**Weaknesses:**

1. Unknown malware detection has been researched for a long time, such as [1]. The proposed method is not a very novel method.

2. Baselines are very limited and only include GCN and GraphSage. (Also, they should called baselines rather than benchmarks). The baselines should cover other OOD methods like one-class SVM.

3. From the visualization in Figure 5, some families cannot be well clustered, which may affect the classification performance.

4. The dataset does not contain the benign software dataset. In practical usage, the model should divide the benign and malicious software.

5. The paper was written in an unprofessional manner: For example, Equation (1), (2), (3), and (5) are the same. The dataset subsection should not be put in the Methods section. All figures have no explanations after the figure title. Training loss figures should not be presented in the main text.

[1] Hashemi H, Azmoodeh A, Hamzeh A, et al. Graph embedding as a new approach for unknown malware detection[J]. Journal of Computer Virology and Hacking Techniques, 2017, 13: 153-166.

**Questions:**

1. What is the concrete family in the testing set and training set?

2. The baselines GCN and GraphSage cannot reject the class (for an unknown family), so the new family accuracy should be 0. Why there is still acc reported in Table 1?

---

> ### Author Response · Authors · 2023-11-23
>
> **Responses to Weaknesses**
> Thank you very much for your comments on our work.
> 1.
> Hashemi et al. implements a graph embedding technique based on OpCodes, representing a lower-level abstraction in capturing malware behavior. In contrast, our work integrates (CFGs) and the centroid model directly into the Graph Neural Network (GNN). Our use of CFGs allows for a higher-level abstraction, capturing more nuanced and semantically rich patterns in malware execution. The utilization of Control Flow Graphs adds a unique layer of sophistication to our approach, enabling the extraction of intricate, context-aware features that are essential for accurate malware classification. Furthermore, our Graph Centroid Model (GCM), embedded within a Graph Neural Network (GNN), dynamically creates representatives in the embedding space. This dynamic nature empowers our model not only to efficiently classify known malware families but also to adapt to emerging threats, making it highly responsive to the evolving landscape of malware.
>
> While CFGs have already shown promise in malware classification, our work introduces a new methodology, Centroid Nets, specifically designed for detecting out-of-distribution (OOD) new-family threats. The CFGs, processed by our Centroid-enabled GNN (GCM), capture intricate patterns, enabling effective classification of known families and the identification of emerging threats. A critical distinction lies in our innovative use of centroid models, providing a dynamic and adaptive mechanism for classifying malware families. Importantly, our workflow for constructing CFGs involves an open-source tool, providing an accessible and cost-free alternative to the often-expensive IDAPro. This democratization of tools is crucial for fostering collaboration and innovation within the academic community, allowing researchers easy access to the analysis of malware families.
>
> We will be revising our paper with distinct and explicit wording to emphasize the novelty of our Centroid model and its contributions. We appreciate the opportunity to clarify these points and further highlight the uniqueness of our approach.
>
> 2. We will ensure that Table 1 includes a more comprehensive comparison with recent approaches for new class family detection, providing a clearer picture of the effectiveness of our algorithm. We will also use center loss in our follow-ups. We will certainly include more baseline analysis in our future version of this work.
>
> 3. Malware datasets often exhibit complexities, including diverse variations within a single family. The visualization serves as a valuable insight into the nuanced nature of malware distribution. Our model is designed to handle such complexities by leveraging the embeddings and centroid-based approach, which, as shown in our experiments, enhances the classification performance, especially in the context of novel and evolving malware families. In fact, Figure 5 may also illustrate that some families are mislabeled and could provide information in malware families that actually contain multiple separate types of malware. This could be a useful feature of our work for malware analysts.
>
> 4. We appreciate the suggestion regarding the inclusion of a benign software dataset. Our focus is primarily on malware family discovery and classification rather than binary detection. In a real-world scenario, our model can be part of a larger cybersecurity system where distinguishing between benign and malicious software is handled by complementary components. However, we understand the importance of a holistic cybersecurity solution, and we hope to explore  malware detection for our update of this work.
>
> 5. We appreciate your feedback on the presentation and formatting of the paper. In the upcoming update, we will address the concerns you raised:  We will remove repetitions of the equation and move the dataset subsection to  outside the Methods section for a more logical structure.  In the revision, we will provide concise and informative captions for all figures, and we will move the training loss figures in the supplementary material rather than the main text to enhance readability.
>
> **Questions**
> 1.  Our dataset consists of CFGs that we retrieved from samples of various malware families, examples which include `berbew` and `sfone`. For each experiment, the training set comprises samples of known families used to train and fine-tune the model. The testing set, on the other hand, contains samples from both known families (for evaluating model performance on familiar threats) and completely new, unseen families (for evaluating the model's capability in detecting novel threats). We ensured a diverse representation of families to provide a comprehensive evaluation of our approach.
> 2. We will remove these two values in our update.

---

### Official Review · Reviewer_TJg9 · 2023-10-30

**Soundness:** 3 good
**Presentation:** 2 fair
**Contribution:** 2 fair
**Rating:** 5
**Confidence:** 3

**Summary:**

The authors propose a novel method for detecting new data classes based on Control Flow Graphs. They employ Graph Neural Network and Centroid Nets to embed the control flow graphs into a latent space. By quantifying the dissimilarities among samples in the embedding space, the authors enable the identification of multiple distinct representations of familiar classes.

**Strengths:**

1.The paper demonstrates clear logic and a well-defined motivation.

2. The research topic is of significant importance.

**Weaknesses:**

1. The design of Centroid Nets is ambiguous.
2. There is a lack of comparison with recent methods for detecting new class families.

**Questions:**

1.	The author did not provide a clear explanation of the training process for Centroid Nets. Additionally, equations (1), (2), (3), and (5) are identical, and these repetitive equations do not offer any additional useful information.
2.	The author should compare the recent approaches for new class family detection in Table 1, as solely comparing with GCN and GraphSAGE does not demonstrate the effectiveness of the algorithm.
3.	In practical usage, there is no separate validation set consisting of samples from new families to determine hyperparameters. In this scenario, how should epsilon be determined?

---

> ### Author Response · Authors · 2023-11-23
>
> Thank you for recognizing the clarity in our paper's logic and the importance of the research topic.
> We acknowledge the concern and agree that clarity is crucial. In the revised manuscript, we will provide a more detailed and explicit explanation of the design of Centroid Nets. This will include a step-by-step breakdown of the model architecture, training process, and proofs of correctness. We have included some of what we will add below.
> We appreciate this feedback. In the revised version, we will include a comprehensive comparison with recent state-of-the-art methods for detecting new class families. This will offer a more thorough evaluation of our approach against other OOD methods, rather than other GNN methods.
> Our use-case, malware datasets, would contain both new and old families in any given validation set. We would tune epsilon to the malware families that we do have in both the training and validation set.
>
> **Algorithm 1: Centroid Partitioning for Malware Family Discovery**
>
> 1. **Input:** `samples` - set of data samples, `threshold` - distance threshold
> 2. **Output:** `centroids` - set of centroids representing malware families
>
> 3. Initialize an empty set of `centroids`.
> 4. Choose a random sample from `samples` and add it to `centroids`.
> 5. **for each** `sample` **in** `samples` **do**
>    1. Compute distances (`distances`) from `sample` to all centroids.
>    2. Find the minimum distance (`min_distance`) from `distances`.
>    3. **if** `min_distance > threshold` **then**
>       1. Add `sample` to the set of `centroids`.
>    4. **end if**
> 6. **end for**
> 7. **return** `centroids`.
>
> **Algorithm 2: Compute Distances**
>
> 1. **Input:** `sample` - a data sample, `centroids` - set of centroids
> 2. **Output:** `distances` - list of distances from `sample` to each centroid
>
> 3. Initialize an empty list `distances`.
> 4. **for each** `centroid` **in** `centroids` **do**
>    1. Compute the distance (`distance`) between `sample` and `centroid`.
>    2. Add `distance` to the list of `distances`.
> 5. **end for**
> 6. **return** `distances`.
>
> The CentroidPartitioning procedure takes in a set of samples and a threshold distance value. It starts by randomly selecting one sample from the set as the first centroid. It then iterates over each sample and computes the distance to all existing centroids using the ComputeDistances procedure. If the minimum distance to any centroid is greater than the threshold, the sample is added as a new centroid.
>
> The ComputeDistances procedure takes in a sample and a set of centroids. It iterates over each centroid and computes the distance between the sample and the centroid using a distance metric such as Euclidean distance or Manhattan distance. It returns a list of distances between the sample and each centroid.
>
> Centroid Proof of Differentiability
>
> To prove the differentiability of $L_j$ with respect to $f(x)$, we will calculate its gradient, $\nabla L_j$.
>
> First, we express $L_j$ in terms of the individual components of $f(x)$ and $c_j$ (where $c_j$ are centroids):
>
> \begin{equation}
> L_j = \sum_{i=1}^m (f(x)_i - c_j)_i^2,
> \end{equation}
>
> where $(f(x)_i - c_j)_i$ denotes the $i$-th component of the vectors $f(x)$ and $c_j$.
>
> Now, we compute the partial derivative of $L_j$ with respect to the $k$-th component of $f(x)$:
>
> \begin{align}
> \frac{\partial L_j}{\partial f(x)_k} &= \frac{\partial}{\partial f(x)_k} \sum_{i=1}^m (f(x)_i - c_j)_i^2 \\
> &= \sum_{i=1}^m \frac{\partial}{\partial f(x)_k} (f(x)_i - c_j)_i^2 \\
> &= 2 \sum_{i=1}^m (f(x)_i - c_j)_i \delta_{ik},
> \end{align}
>
> where $\delta_{ik}$ is the Identity matrix, which is 1 when $i = k$ and 0 otherwise.
>
> We combine these partial derivatives to form the gradient vector $\nabla L_j$:
>
> \begin{align}
> \nabla L_j &= \left(\frac{\partial L_j}{\partial f(x)_1}, \frac{\partial L_j}{\partial f(x)_2}, \ldots, \frac{\partial L_j}{\partial f(x)_m}\right) \\
> &= 2 \left((f(x)_1 - c_j)_1, (f(x)_2 - c_j)_2, \ldots, (f(x)_m - c_j)_m\right) \\
> &= 2 \left(f(x)_1 - c_j)_1, f(x)_2 - c_j)_2, \ldots, (f(x)_m - c_j)_m\right) \\
> &= 2 (f(x) - c_j).
> \end{align}
>
> Therefore, the gradient $\nabla L_j$ of $L_j$ with respect to $f(x)$ is $2 (f(x) - c_j)$.

---

### Meta-Review · Area_Chair_BA28 · 2023-11-30

**Metareview:**

This paper employs a space partitioning strategy to malware classification and detection of new malware classes, unobserved during training. A novel dataset is further introduced. While reviewers consistently highlighted the relevance of the topic, they agreed on limitations that require improvement prior to publication. Specifically, concerns with the presentation, the evaluation, and novelty were highlighted.

The paper could benefit from improvements in the presentation including the addition of details about the model and training instead of learning curves that could be part of the appendix. The evaluation should also cover existing methods able to perform selective classification and reject instances from new classes. I'd also suggest accounting for recent work showing evidence that standard classifiers perform selective classification as well as or better than other recent methods. In terms of motivation, I'd suggest for the authors to better justify their focus on malware detection since their proposal can be applied in any multi-class classification setting.

**Justification For Why Not Higher Score:**

While the paper is quite interesting, there are multiple issues that should be addressed before being ready for publication as outlined in the latter portion of the meta-review.

**Justification For Why Not Lower Score:**

N/A

---

### Decision · Program_Chairs · 2024-01-16

Reject